# The Affect Misattribution in the Interpretation of Ambiguous Stimuli in Terms of Warmth vs. Competence: Behavioral Phenomenon and Its Neural Correlates

**DOI:** 10.3390/brainsci12081093

**Published:** 2022-08-17

**Authors:** Kamil K. Imbir, Joanna Duda-Goławska, Gabriela Jurkiewicz, Maciej Pastwa, Adam Sobieszek, Adrianna Wielgopolan, Jarosław Żygierewicz

**Affiliations:** 1Faculty of Psychology, University of Warsaw, 00-183 Warsaw, Poland; 2Biomedical Physics Division, Institute of Experimental Physics, Faculty of Physics, University of Warsaw, 02-093 Warsaw, Poland

**Keywords:** emotional words, valence, origin, ambiguous task, response-locked ERPs

## Abstract

Warmth and competence are fundamental dimensions of social cognition. This also applies to the interpretation of ambiguous symbolic stimuli in terms of their relation to warmth or competence. The affective state of an individual may affect the way people interpret the neutral stimuli in the environment. As previous findings have shown, it is possible to alter the perception of neutral social stimuli in terms of warmth vs. competence by eliciting an incidental affect with the use of emotion-laden words. In the current experiment, we expected the valence and origin of an affective state, factors ascribing emotionally laden words, to be able to switch the interpretation of the neutral objects. We have shown in behavioural results that negative valence and reflective origins promote the interpretation of unknown objects in terms of competence rather than warmth. Furthermore, electrophysiological-response-locked analyses revealed differences specific to negative valence while making the decision in the ambiguous task and while executing it. The results of the current experiment show that the usage of warmth and competence in social cognition is susceptible to affective state manipulation. In addition, the results are coherent with the evolutionary perspective on social cognition (valence effects) as well as with predictions of the dual mind model of emotion (origin effects).

## 1. Introduction

The interaction between emotions and cognition is an important issue for understanding the decision-making process [1,2,3]. Whether the emotional context can shape the outcomes of decisions is especially interesting, but so is the process of decision making itself. In the current experiment, we examined the role of valence and the origin of emotion [1,4,5,6] in decision making involving ambiguous stimuli interpretations [7,8]. Recent behavioral studies have shown that valence (positive vs. negative) and origin (automatic vs. reflective) can shape the interpretation of ambiguous stimuli in terms of their warmth and competence [9]. In the current experiment, we employed event-related potentials (ERPs) to investigate the time course of the decision-making process and the role of an incidental affect accompanying the process.

### 1.1. Warmth and Competence in Social Cognition

Warmth and competence are thought to be fundamental dimensions of social cognition [10,11], allowing us to think about the social environment, interpret it, make some heuristic assumptions and accordingly plan our behaviors [12,13]. These dimensions may have different names denoted to them, starting from the aforementioned warmth and competence, communion and agency [10] or expressiveness and instrumentality. Nevertheless, they underlie the social judgements itself [13] and remain crucial in this process [10]. The first of them, warmth, is related to other people’s assumed intentions towards us. With “warmth” (communion, expressiveness), there are associated traits such as trustworthiness, helpfulness, honesty and sociability—traits that are, in general, rather positive and related to being good-natured. The opposite traits, which are perceived as negative, are associated with “cold” people, described as unpleasant and deceitful [14]. The dimension of competence (agency, instrumentality) describes people’s abilities to realize their intentions. Being high on the competence dimension is associated with positive traits such as cleverness, creativity, efficiency, fore-sighting, intelligence and knowledgeability [11,15], whereas people that are low on this dimension are described as lazy, inefficient and passive [14]. Such presented dimensions create a space for the interpretation and understanding of other people’s actions in many different aspects of social functioning [10,16], accounting for approach-avoidance tendencies [17], quick interpretations of unknown faces in terms of warmth but not competence [18] as well as inferences about the motives of other people [19].

### 1.2. Ambiguous Stimuli Interpretation Paradigm and the Role of Emotional Factors

People deal with ambiguity on a daily basis in regular decision-making processes, engaging their mind into evaluating possible answers and finding the right one in their opinion [20]. The cognitive processing of a task with at least two possible answers (also called open tasks) requires participants to deal with this specific kind of ambiguity in the sense of decision making [21]. The possibility of different answers alone creates a space for interpretations and allows for an individual choice; these circumstances may be even more pronounced if the answer is not obvious and if there are no additional clues [20]. This interpretation and the final outcome—the decision made by the participant—may be susceptible to mindset manipulations [21] and emotion induction [22], among which the use of emotional characteristics of processed word stimuli stands out as particularly effective [23]. In the current experiment, we were especially interested in investigating the impact of the valence and origin of emotion on decisions concerning ambiguous stimuli. The dimension of valence refers to the pleasantness vs. unpleasantness of the stimuli [24] and is considered one of the most basic dimensions underlying emotional experiences (a crucial part of the core affect theory) [24,25], shaping its nature and, as a consequence, the attitude or behavior of an individual [26]. In our study, we decided to distinguish three levels of valence: negative, neutral and positive [27].

The origin is a dimension related to the process of the formation of the emotion, which is either automatic or reflective [6,28]. Emotions of automatic origin appear immediately after stimuli onset and do not require much of a conscious effort. They are usually based on biological criteria (often derived from evolutionary processes), which are intuitive and hard to verbalize [6,29]. On the other hand, emotions originating from reflective processing require more time and effort; their appraisal criteria are easier to verbalize, usually derived from some cognitive processes, already obtained knowledge and deliberative comparisons of the actual situation with the model of an ideal one [8].

### 1.3. The Link between Affect and Social Cognition

Both aforementioned emotional dimensions of warmth and competence may be crucial in the process of evaluating stimuli (e.g., [1]) and relevant to the dual process theories, assuming the existence of two complementary processing modes based accordingly on experimental and rational processing [28,30]. These two modes are very well established in cognitive psychology; however, it seems that there can be a duality of processing also for the specifics of affective functioning on different dimensions [1,6]. For example, previous studies have found that the positive valence significantly increases the assessments on both dimensions of warmth and competence [31], whereas the origin has a dual influence—namely, the automatic origin was promoting higher assessments for warmth, and the reflective origin was causing higher assessments for competence [8,31]. Furthermore, the origin dimension alone influences the decision in an ambiguous task, with reflective words resulting in significantly longer reaction times than the automatic or unspecific origin [23]. These results may be a manifestation of two separate systems, which can also interlace and shape the perception of social dimensions (warmth and competence) influenced by emotional dimensions (valence and origin).

### 1.4. EEG Correlates

#### 1.4.1. Event-Related Potential Components Related to Word Processing

One of the reasons word stimuli are valuable tools in the study of the interaction between emotions and decision making is that the processing of words is an automatic process with a well-defined time-course that can be investigated with the use of event-related potentials (ERPs; [32]) Two distinctions are instructive in the understanding of electrophysiological correlates of affective word processing. The first distinction is concerned with the type of task that participants are engaged in when reading. The second distinction can be made within the process of word-reading itself and is concerned with temporally distinct stages of word processing. Both these distinctions are informative, as a different emotional modulation is expected in each processing stage based on the type of task the participants are engaged in [32,33].

The processing of emotion-laden words can be characterized in three stages [34,35]. The first stage is a visual analysis of the word that begins from stimulus onset and lasts for about 200 ms [35]. There, we observe the effects of the word’s orthographic features, as well as the familiarity effects (e.g., at the P150 component; [36,37]). The second phase is associated with an automatic processing of the word and can be identified by the onset of emotional modulation, which starts between 200 and 300 ms after onset [32]. This stage is associated with modulation in early components, such as the early posterior negativity (EPN). The last stage, associated with later ERP components such as the late positive complex (LPC), whose amplitude peaks between 500 and 800 ms, reflects the attention capture associated with the word, as well as a deeper assessment of its meaning [38], which may be needed to respond to the task at hand. However, the existence and pattern of emotional modulation at this stage is greatly influenced by the type of task that the participant is engaged in [39] and occurs mostly when the execution of the task requires such a deliberate assessment [40,41].

Thus, the second distinction specifies the type of task being performed by the participants. In the study of emotional word reading, we can identify two broad approaches dealing with involuntary or voluntary processing [32,33]. The first employs tasks such as the emotional Stroop task [42,43,44], where a deeper processing of the word’s meaning is not required and where we observe modulation occurring mostly in early components [45,46]. The second type employs tasks that require an explicit, deeper processing of words, such as the emotional decision task [47], where we observe modulation on later components, such as the LPC.

In our experiment, we employed a silent word reading task, where participants had to (1) memorize the words, (2) interpret an ambiguous stimulus and (3) recall the words in order. Although silent word reading itself is a task that does not require a deeper processing of the word and, therefore, most such tasks only produce an early, non-significant EPR response, tasks involving memorization and working memory are an exception [46]. For example, in a study where participants were instructed to try to remember 180 displayed words as much as possible, early modulation was observed for stimuli differing in valence [46]. Differences in the amplitude of around 250 ms after stimulus onset were found, with both positive and negative stimuli eliciting a more negative amplitude when compared to neutral stimuli. This modulation was observed especially at occipital electrodes. In a further investigation the differences in the N1 and EPN components were found, caused by the emotionality of the word stimuli and thus showing how the processing of valence in linguistic stimuli may be mostly found in early components [48].

The modulation expected in a task of remembering a word cannot, however, be compared to a typical voluntary (deeper) processing task for two reasons. First, such a task lacks a need for an immediate response, a characteristic for tasks where we observe a late modulation. Second, although a late modulation may still occur if participants engage in a semantic analysis of the word’s meaning, this does not necessarily occur in such a task. This is because an individual may employ different strategies to remember a word, most of which do not require deep semantic processing of the word’s meaning [49]. Rather, in such a paradigm, we should understand the participant’s task as storing the word in working memory. Working memory is the system that allows for temporary storage of information as to manipulate or act upon it [50]. The conceptual and lexical working memory index is the N400 component [51,52], which reflects the lexical integration processes. Typically, new information from sensory inputs is compared to information stored in working memory, and an inconsistency elicits a greater negativity exactly at the N400 time range [53]. N400-like component potentials have also been obtained in studies using a recollection paradigm. Previous research has shown N400 to be susceptible to both the valence and origin of the word [54]. Specifically, in the 350 to 500 ms time range, there was a change in amplitude in parietal regions for positive and negative words (the effect of valence), and the emotional dimension of the origin shaped amplitudes in the frontal regions.

#### 1.4.2. Visual Processing of Ambiguous Stimuli

During the perception of ambiguous stimuli, several factors may be involved in modulating the stimuli processing, as indexed by the ERP amplitudes. In our study, we may expect modulation due to different task demands, as well as emotional modulation caused by holding emotional words in memory. Two means by which we may expect this influence to occur are through an influence on the ERPs associated with attentional processes involved in the evaluation of visual stimuli, as well as through the modulation of the particular ERPs associated with stimuli ambiguity.

It has been reported that two ERP markers of stimuli ambiguity occur 200 ms and 400 ms after stimuli onset. For visually ambiguous stimuli, such as the Necker cube, amplitude differences in the anterior P2, as well as the posterior P400, have been reported when compared to disambiguated versions of the same stimuli [55,56], with ambiguous stimuli producing smaller amplitudes. Visual stimuli may be ambiguous in different ways depending on the dimension on which it is evaluated. Nonetheless, remarkable similarities in the ERP markers have also been shown for ambiguity of motion and semantic content ambiguity, such as with Boring’s ambiguous Old/Young Woman picture, which led Kornmeier et al. [56] to deem these components markers of more general “ERP Ambiguity Effects”. Joos et al. [57] further generalized these results by obtaining similar results for tasks of comparing ambiguous emotional expressions, as well as minimally different abstract figures.

Minimal EEG research has been conducted specifically on the interplay between emotional priming and the visual processing of ambiguous stimuli. However, effects of emotions on the attentional process during visual processing have been reported, e.g., the influence of positive mood on early (100 ms after onset) processing in the V1 region [58]. Studies of mood effects on perceptions of emotional intensity have also revealed modulation in the P3 component time range. Cavanagh and Geisler [59] used facial stimuli with depressed and non-depressed individuals and showed how depressed individuals had reduced P3 amplitudes for happy faces. Interestingly, the effect occurred only for happy faces with a low intensity of the emotional expression, showing that the effect may be due to the processing of ambiguous, mood-incongruous stimuli.

#### 1.4.3. ERP Components Related to Deciding

The potentials evoked by the stimuli may not only be related to processing the displayed stimulus but also to the actual response that the participant should make in a particular task. Neural activity signalizing the preparation for response is called lateralized readiness potential (LRN; [60,61]). The LRN is observed before the hand movement (e.g., pressing a key in response to the stimuli), over the parietal site of the scalp and contralateral to the hand that is being used for the response [62]. For example, if the right hand is used to answer in the task, the LRN would be observed in the potential recorded on the left side of the scalp, by the C3 electrode (in the 10–20 system). The LRP is commonly extracted in a response-locked manner, which is opposite to the stimulus-locked manner used for most other components [62,63]. The time before the peak amplitude of the LRN is tied to the generation of the response to particular stimuli, whereas the time from the peak of the amplitude to the response is tied to the execution of the response [63,64].

The LRN potential is sensitive to the cues that are displayed before the response should be made [65]. The cues especially influence the part before the peak amplitude related to the preparation for the response, and the response execution is rather influenced by the complexity of the task itself [62]. Depending on the cues, the participant could prepare for a particular reaction, which could be seen in the LRN signal, even if the response was not executed. The cues could be purely cognitive, signalizing, for example, which hand should be used to react in the task [65,66]. Some studies suggest that the LRP could also be affected by emotional stimuli, which, in this case, are treated as certain cues about the response that should be executed. Results of experiments using emotionally laden pictures as the stimuli have shown that the LRP was influenced by the emotional stimuli in both the processing and execution phase; however, only the negative stimuli affected the execution process [67,68].

In our previous research, we also found that the emotional stimuli influence the LRP amplitudes. Emotionally laden words were affecting the LRP in an ambiguous task in which the participants were asked to pick which of the two randomly generated QR codes better reflected the meaning of a certain word. The words were differing in valence (negative, neutral or positive) and origin (automatic, no particular origin or reflective) levels; however, we found only the effects of the origin to be significant in this task, with reflective words evoking more positive amplitudes. The difference between the signals observed throughout both the processing and execution phase was particularly interesting; however, we identified the lateralization of the signal only in the execution phase [54]. This indicates that the LRP is sensitive to emotion-laden words in tasks requiring decisions; thus, we expected differences in this component in the study presented in this article.

### 1.5. Aim and Predictions

The aims of the current EEG experiment based on the ambiguous stimuli interpretation paradigm were: (1) to investigate the role of valence and origin on decisions concerning warmth and competence treated as separate scales and (2) to investigate neural correlates of ambiguity processing under the influence of affectively charged words stored in memory during the task execution. As experimental manipulation, we involved the presentation of words differing in valence (3 levels) and origin (2 levels).

In accordance with the emotion duality model [1,69], we expected dissociative effects on the behavioral level, i.e., that automatic originated emotions would promote the interpretation of ambiguous stimuli in terms of warmth but not competence, whereas reflective originated emotions would promote the interpretation in terms of competence but not warmth.

On the neurobiological level, we expected to find two types of effects in ERP amplitudes related to the valence and origin of emotional stimuli, as well as the dimension of social cognition evaluated: (1) for silent reading and remembering the emotion-laden word, (2) for the visual inspection of ambiguous stimuli and (3) for decision making in an ambiguous task [54]. Due to the fact that this was the first experiment investigating the neural correlates of a behavioral phenomenon, we decided on an exploratory approach in EEG analysis. In silent word reading, we expected to find differences for the valence and origin of emotional stimuli. In the visual inspection of stimuli and decision making, we expected to identify mostly the impact of the social cognition dimension. The rationale for such an expectation is that the task was directly linked to the interpretation of stimuli in terms of warmth vs. competence. In the context of decision making, the LRP was especially interesting for us—the measure indexed processes preceding the decisions, thus integrating both emotional and social cognition factors of our design.

## 2. Methods

### 2.1. Participants

We conducted an a priori estimation of our sample size. Basing on previous studies, we assumed that the η_p_^2^ for the main effects of one factor would be about 0.10. The estimations using G-Power software (version 3.1.9.4., created by Franz Faul, Universitat Kiel, Kiel, Germany) [70] showed that, to achieve the statistical power of *α* = 0.80, we would need at least 20 participants. That small sample size is dictated by the specific study design—a significant number of repeated measures trials. However, we decided to increase the sample up to 36 participants in order to be able to conduct reliable interactions analyses and to have a possibility to exclude any potential outliers.

The participants were recruited from various faculties of Warsaw universities. The inclusion criteria were: right-handed native Polish speakers without chronic clinical issues that may affect EEG recording directly (e.g., epilepsy) or because of the medication prescribed because of this illness and with normal or corrected-to-normal vision. They received a small payment (about 20 Euros) for taking part in the experiment.

The entire experimental group consisted of 36 subjects (50% men and 50% women) aged from 19 to 28 years old (*M* = 23.00; *SD* = 2.32). After collecting the data, some participants were excluded from certain parts of the EEG analyses because of excessive artifacts or extremely short or long response times. Namely, there were 28 participants included in the analysis of the word reading (14 men and 14 women, aged 19–28 years, *M* = 22.75, *SD* = 2.05); 30 participants in the analysis of the hexagram viewing section (15 men and 15 women, aged 19–28 years, *M* = 23.07, *SD* = 2.20); and 26 participants in the analysis of the decisions concerning hexagrams, with both EEG and behavioral data (13 men and 13 women, aged 19–28 years, *M* = 23.00, *SD* = 2.21).

The participants provided informed consent to participate in the experiment, and the fact was documented in the research diary. The bioethical committee approved the design, experimental conditions and procedure. All of the procedures involving human participants followed the ethical standards of the institutional and national research committee and the 1964 Declaration of Helsinki and its later amendments or comparable ethical standards.

### 2.2. Design and Statistical Procedures

We investigated the behavioral and electrophysiological correlates of the execution of an ambiguous task under the influence of the emotionality of words stored in the working memory. We manipulated the levels of valence (three levels: Neg—negative, Neu—neutral and Pos—positive) and their emotional origin (two levels: Auto—automatic and Refl—reflective), ensuring that stimuli were matched for arousal, concreteness, frequency of appearance in language, and length. The ambiguous task was to answer one of the two questions: How warm or competent is an octagram? Thus, it created a two-level factor “type of question”. Furthermore, we used two lists of words in this study (list A and B; see Figure 1); list A was our experimental stimuli (diversed by valence and origin, as we mentioned before), whereas list B was the buffer list (diversed by other emotional dimensions, arousal and subjective significance, and thus irrelevant for this study). Those two lists were not interlaced with each other, and we only analyzed the described list A, as it was the one within the scope of the present study.

We used a three-way analysis of variance with repeated measures (valence × origin × type of question) in analyzing the behavioral effects. We investigated the interaction effects occurring between the factors at subsequent steps through the analysis of variance, which took the interacting factors from a previous step as independent variables. We continued the analysis to a level at which one could understand the interactions in terms of differences in the effects of simple factors or by the interaction of two factors under specific conditions determined by the particular levels of the other factor. We performed the post hoc using pairwise *t*-tests. We handled the problem of multiple comparisons utilizing the Holm procedure [71]. We checked the sphericity with Mauchly’s test and applied the Greenhouse–Geisser correction where necessary.

In the case of EEG correlates, we were likewise interested in the spatial distribution of the effects. Therefore, we included a fourth factor—regions of interest (ROI). ROI had four levels in the case of the stimulus-onset-aligned analysis and nine levels in the decision-aligned analysis. Therefore, the EEG part of the experiment had a four-factor design. The mean component amplitude within a given time window was the dependent variable, and the independent variables were valence, origin, type of question and ROI. We used a hierarchical analysis of variance to examine the effects. At the first level, we performed a four-factor analysis of variance with repeated measures within each time window separately but took into account the correction of the level of significance by the Holm procedure for corresponding effects in the consecutive windows, and we continued analogously for the behavioral data. The procedures were implemented in the R statistical package [72].

### 2.3. Materials

#### 2.3.1. Properties of Words

The words used as stimuli in our experiment were taken from the Affective Norms for Polish Words Reload database [27]. In this database, normative values were assigned to 4900 Polish words based on the participants’ assessments. The assessments for each word were performed by at least 50 participants, with men and women in equal proportion. We chose words with extreme values on the scales of valence and origin to create groups of experimental stimuli, divided as follows: 3 levels of valence (negative, neutral and positive) and 2 levels of origin (automatic and reflective), i.e., 6 groups total. There were 15 words in each group. We assumed that the variation of stimuli on other dimensions should be controlled, as it could have influenced the processing of words. The controlled dimensions were: concreteness, arousal, the frequency of usage in the Polish language [73] and the length of the word (number of letters).

We conducted two-way ANOVA analyses in order to verify the division of stimuli. The groups divided by the factor of valence differed significantly on the dimension valence (*F*(2, 84) = 413.31; *p* < 0.001; *η*^2^ = 0.92); they did not differ on the dimension of origin (*F*(2, 84) = 0.80; *p* = 0.46; *η*^2^ = 0.02), nor on any controlled dimension. The groups divided by the factor of origin differed on the dimension of origin (*t*(88) = −20.33; *p* < 0.001; *d* = 4.29), whereas there were no differences on the dimension of valence (*t*(88) = −0.27; *p* = 0.79; *d* = 0.06), nor any other controlled dimension. The complete list of used words with their normative values, as well as the means from normative values for word groups and analyses of differences between them on controlled dimensions, may be found in Appendix A.

#### 2.3.2. Ambiguous Task

Open tasks, which do not have one specific correct answer or have a whole spectrum of possible correct answers, require ambiguity processing [20]. Such ambiguous tasks are particularly attractive to researchers, as the answer may be vulnerable to the manipulation of the mindset or emotional state [21,22]. Previous research has shown that the emotional properties of words related to valence and origin (automatic vs. reflective) affect the processing of an open, ambiguous task [23]. In the reflective condition, the latencies of processing the task were more prolonged than in the automatic condition, which led to the conclusion that the origin of emotion promoted a congruous way of processing the task.

An ambiguous task employed for this experiment was based on assessing a symbol without a specific meaning on one of two scales—warmth or competence, the previously described fundamental dimensions of social cognition. Assessing culturally non-specific symbols is a procedure commonly used for evoking ambiguity [74,75]. The stimuli used in our study were octagrams. An octagram is a square graphical symbol constructed of eight horizontal lines separated by white spaces. The top and bottom lines are always solid and black. The inner lines have three equally distant intervals that can be white or black.

### 2.4. Procedure

The subject’s task was to assess the displayed octagram on one of the two fundamental dimensions of social cognition: warmth or competence. In a given trial, the exact question was either “On a scale from 1 to 4, how much does this symbol represent the idea of warmth?” or “On a scale from 1 to 4, how much does this symbol represent the idea of competence?”. The scale on which the participant evaluated the octagram in a given trial defined the experimental condition “type of question”. We assumed that these two dimensions (dimension of competence or warmth) would not be related in any way to the symbols, as the octagrams did not have any actual meaning and were generated randomly.

The words were presented in blocks. In one block, 90 stimuli (3 levels of valence × 2 levels of origin × 15 words for each combination of levels) were displayed randomly. At the beginning of each block, a subject was shown instructions telling him or her which properties of the octagram should be evaluated for warmth or competence. The first property to be evaluated was randomly assigned for each subject. Successive blocks had alternating evaluations. Altogether, the experimental session had four blocks. Each block was followed by a 3 s pause for the eyes to rest. After two blocks, the subject had time to rest for as long as needed. 

The sequence of a single trial was as follows: a fixation crosshair sign was displayed (550 ± 50 ms), followed by a word to be memorized (visible for 900 ms). After the word faded, the screen remained empty for 300 ms, and then the fixation crosshair was displayed again (550 ± 50 ms). Next, an octagram was shown on the screen. The octagram was displayed until the subject indicated the level of warmth or competence by pressing the appropriate key on the keyboard; the octagram sign was randomly assigned to each word within a given stimulus list presentation. After the subject’s response, the screen went blank for 300 ms, followed by the display of a cross sign (550 ± 50 ms). Next, a pair of words was displayed. The subject had to indicate, by pressing the appropriate key, which of the words was displayed at the beginning of the test. This part of the trial was intended to control the subject’s attention. The whole experimental procedure is presented in Figure 1.

### 2.5. EEG Recording

#### 2.5.1. Apparatus

The stimuli were displayed on a standard personal computer monitor (LCD; 15-inch diagonal). The stimuli were synchronized to EEG data utilizing a circuit that recorded the changes in the brightness of a small rectangle on the display, covered by a detector and therefore invisible for the subjects. Its brightness varied synchronically with the content of the screen. We recorded EEG signals from 19 electrode sites; Fz, Cz, Pz, Fp1/2, F7/8, F3/4, T7/8, C3/4, P7/8, P3/4 and O1/2 referenced to linked earlobes. The ground electrode was placed at the AFz position. All impedances were kept at a similar value below 5 kΩ. The signal was acquired using a Porti7 (Tmsi) amplifier sampled at 1024 Hz.

#### 2.5.2. Offline EEG Processing

We conducted the offline signal processing utilizing Matlab^®^ (version 9.2.0.538062 R2017a) with the EEGLab toolbox (v2019.1, created by Delorme and Makeig, University of California, San Diego, CA, USA) [76] and custom-made scripts. The signal was zero-phase filtered (filtfilt procedure). We used the second-order Butterworth filters with 12 dB/octave roll-off; the high-pass filter cutoff was 0.1 Hz, and the low-pass cutoff was 30 Hz. We also used the notch filter for the 49.5–50.5 Hz band, which was also implemented as the second-order Butterworth filter.

We extracted three types of epochs for further analysis. Two of them were aligned to the word and octagram onsets, respectively. In both cases, we extracted intervals ranging from −100 to 600 ms. The signals were baseline corrected to the period −100 to 0 ms. After removing the trials with artifacts (based on the analysis of the abnormal values and trends, followed by visual inspection), there were *M* = 22.8, *SEM* = 0.3 epochs per condition in the epochs aligned to word onset and *M* = 22.8, *SEM* = 0.2 epochs aligned to the octagram onset.

The third type of epochs ranged from −600 to 0 ms preceding the subject’s key press, communicating the evaluation of the octagram. Here, we performed the baseline correction to the −600–500 ms period. We removed from further analysis trials that contained eye blinks in the analyzed epochs or trials in which the subject did not correctly recall the word presented before the octagram (7.4%). Additionally, we removed trials with response times shorter than the 5th or longer than the 95th percentile of the response time of all the cohorts. Effectively, the response time for the analyzed data was within 287–4962 ms. There were *M* = 21.9, *SEM* = 0.3 epochs per condition.

## 3. Results

### 3.1. Behavioral Results

#### 3.1.1. Recalling Effectiveness

The average recalling effectiveness was *M* = 95.25%, *SEM* = 0.69%. We did not obtain any significant main effects (for valence *F*(2, 50) = 0.21, *p* = 0.8, for origin *F*(1, 25) = 0.56, *p* = 0.4, for type of question *F*(1, 25) = 0.11, *p* = 0.7) or interactions.

#### 3.1.2. Decision Response Times

The average response time was *M* = 1596.0 ms, *SEM* = 141 ms. We did not obtain any significant main effects (for valence *F*(2, 50) = 0.55, *p* = 0.50, for origin *F*(1, 25) = 0.27 *p* = 0.60, for type of question *F*(1, 25) = 0.69, *p* = 0.40) or interactions.

#### 3.1.3. Assessments of Warmth or Competence

We conducted a three-way analysis of variance with repeated measures in which we considered the assessment of octagrams, i.e., the value from 1 to 4 assigned by a subject as a response to the question of “How warm?” or “How competent?” is the symbol, as the dependent variable. The independent variables were valence, emotional origin and the type of assessment (warmth or competence). 

For each of the independent variables, we obtained significant main effects. Namely, for valence, the statistics were *F*(1.13, 28.25) = 15.05, *p* < 0.001, *η**_p_**^2^* = 0.38. The Mauchly’s test indicated that the assumption of sphericity had been violated for this factor (χ^2^(50) = 0.23, *p* < 0.001); therefore, the degrees of freedom were corrected using the Greenhouse–Geisser estimates of sphericity (*ε* = 0.56). The post hoc tests showed that the assessment increased with the valence level. It was the lowest for negative stimuli (*M* = 2.17, *SEM* = 0.85), moderate for neutral stimuli (*M* = 2.36, *SEM* = 0.07) and the highest for positive ones (*M* = 2.63, *SEM* = 0.07). The differences between each level of valence were significant, i.e., negative vs. neutral: *t*(25) = −3.46, *p* = 0.002, *d* = 1.36, neutral vs. positive: *t*(25) = −3.89, *p* < 0.001, *d* = 1.53 and negative vs. positive: *t*(25) = −4.04, *p* < 0.001, *d* = 1.59. These relations are presented in Figure 2a.

We also obtained the main effect of emotional origin (*F*(1, 25) = 13.40, *p* < 0.001, *η**_p_**^2^* = 0.35). The mean assessment for stimuli inducing emotions of automatic origin (*M* = 2.31, *SEM* = 0.06) was lower than that for stimuli inducing reflective ones (*M* = 2.46, *SEM* = 0.06), (cf. Figure 2b).

Finally, we observed a significant effect of the type of question (*F*(1, 25) = 15.98, *p* < 0.001, *η**_p_**^2^* = 0.39). Here, the mean assessment was lower for the question “How warm?” (*M* = 2.31, *SEM* = 0.06) than for “How competent?” (*M* = 2.46, *SEM* = 0.05) (cf. Figure 2c).

Furthermore, we obtained a statistically significant interaction between emotional origin and the type of question (*F*(1, 25) = 9.01, *p* = 0.006, *η**_p_**^2^* = 0.27). Post hoc tests showed that, in the case of the question about competence, the assessment in the automatic condition (*M* = 2.29, *SEM* = 0.08) was significantly lower than that in the reflective condition (*M* = 2.64, *SEM* = 0.07, *t*(25) = −3.64, *p* < 0.001, *d* = 1.43). Moreover, the assessment for reflective words was significantly lower for the question about warmth (*M* = 2.28, *SEM* = 0.08) than that about competence (*t*(25) = −3.80, *p* < 0.001, *d* = 1.49). The assessment for the question about warmth in the automatic condition was (*M* = 2.34, *SEM* = 0.05) and did not differ significantly from the assessments in other conditions. These results are visualized in Figure 2d.

Moreover, the interaction between the type of question and valence was significant (*F*(2, 50) = 4.48, *p* = 0.02, *η**_p_**^2^* = 0.15). The post hoc test showed that, for each type of question, the assessments increased with the level of valence and differed for each pair of valence levels. Namely, for the question “How warm?”, we obtained the following results: negative (*M* = 2.05, *SEM* = 0.1) vs. neutral (*M* = 2.27, *SEM* = 0.08; *t*(25) = −3.71, *p* = 0.002, *d* = 1.46), neutral vs. positive: (*M* = 2.62, SEM = 0.08; *t*(25) = −3.78, *p* = 0.002, *d* = 1.48) and negative vs. positive: (*t*(25) = −4.21, *p* < 0.001, *d* = 1.65). For the question “How competent?”, we obtained the following scores: negative (*M* = 2.28, *SEM* = 0.08) vs. neutral: (*M* = 2.44, *SEM* = 0.07; *t*(25) = −2.25, *p* = 0.03, *d* = 0.88), neutral vs. positive: (*M* = 2.65, *SEM* = 0.07; *t*(25) = −3.06, *p* = 0.01, *d* = 1.20) and negative vs. positive: *t*(25) = −3.45, *p* = 0.005, *d* = 1.35). Furthermore, the assessments within the negative condition were significantly lower for warmth than for competence (*t*(25) = −4.36, *p* < 0.001, *d* = 1.71) and were analogous for the neutral condition (*t*(25) = −2.65, *p* = 0.02, *d* = 1.04). We can summarize this interaction in the following way: for both types of questions, the assessment increased with the level of valence and reached the same value for positive words, but for the question about warmth, these changes were bigger. This interaction is visualized in Figure 2e.

### 3.2. EEG Results

#### 3.2.1. Word Reading and Remembering Task

##### Selection of Time Windows and Regions of Interest

The ERP amplitude was analyzed in four time windows: 60–145 ms, 145–235 ms, 235–350 ms and 350–500 ms. The selection of these ranges was based on the global field power curve (GFP). GFP corresponds to the spatial standard deviation at a given time point in selected time windows. It quantifies the sum of electrical activity over all electrodes [77,78]. GFP maxima were used to determine the latencies of evoked potential components (Figure 3). Microstates corresponding to the components are illustrated in the topographic plots of the amplitude distribution at the bottom of Figure 3. To investigate the possible fronto-posterior and hemispheric contrasts, we selected four regions of interest (ROI): left-frontal (**LF**: Fp1, F3, F7), right-frontal (**RF**: Fp2, F4, F8), left-parietal (**LP**: P3, P7, O1) and right-parietal (**RP**: P4, P8, O2).

##### Results for Word Reading and Remembering Task

We conducted an ANOVA with repeated measures for each of the time windows. The complete statistical results, presenting ANOVA tables for all investigated factors and their interactions, are reported in Appendix A. Here, we describe only the statistically significant results.

Statistically significant effects were obtained only in the time window 235–350 ms, i.e., an interaction between the origin and regions of interest (*F*(1.76, 47.45) = 5.57, *p* = 0.002, *η**_p_**^2^* = 0.18) was observed. The Mauchly’s test indicated that the assumption of sphericity had been violated (χ^2^(81) = 0.28, *p* < 0.001); therefore, the degrees of freedom were corrected using the Greenhouse–Geisser estimates of sphericity (ε = 0.59). Further analyses within ROIs revealed a statistically significant main effect related to the levels of origin in **LP** (*F*(1, 27) = 12.13, *p* = 0.002, *η**_p_**^2^* = 0.31) and **RP** (*F*(1, 27) = 15.36, *p* = 0.002, *η**_p_**^2^* = 0.36). Post hoc tests showed that, in both regions, the amplitude in the reflective condition was more positive than in the automatic one. Namely, we obtained in **LP** reflective (*M* = 2.34, *SEM =* 0.50) vs. automatic (*M* = 1.83, *SEM =* 0.47; *t*(27) = 3.48, *p* = 0.002, *d* = 1.32) and in **RP** reflective (*M* = 2.70, *SEM =* 0.43) vs. automatic (*M* = 2.16, *SEM =* 0.39; *t*(27) = 3.92, *p* < 0.001, *d* = 1.48). The results are presented in Figure 4.

#### 3.2.2. Stimulus-Locked Effects for Warmth–Competence Interpretation of Octagrams

##### Selection of Time Windows

In this part of the experiment, the ERP amplitude was analyzed in four time windows: 55–135 ms, 135–175 ms, 175–315 ms and 315–500 ms. Again, the selection of these ranges was based on the global field power curve (GFP) (Figure 5). We used the same four regions of interest as in the previous section.

In each time window, we performed a four-way analysis of variance with repeated measures. The complete statistical results, presenting ANOVA tables for all investigated factors and their interactions, are reported in Appendix A. Here, we describe only the statistically significant results. We observed significant effects in three time intervals: 55–135 ms, 135–175 ms and 315–500 ms. The details are given below.

##### Time Windows 55–135 ms and 135–175 ms

In both time windows 55–135 ms and 135–175 ms, we observed a statistically significant interaction between the type of question and ROIs. Namely, in the interval 55–135 ms, we obtained (*F*(1.79, 52.02) = 4.95, *p* = 0.01, *η**_p_**^2^* = 0.15). The Mauchly’s test indicated that the assumption of sphericity had been violated (χ^2^(87) = 0.301, *p* < 0.001); therefore, the degrees of freedom were corrected using the Greenhouse–Geisser estimates of sphericity (ε = 0.59). In the interval 135–175 ms, we obtained (*F*(1.38, 40.03) = 6.23, *p* = 0.01, *η**_p_**^2^* = 0.18). The Mauchly’s test indicated that the assumption of sphericity had been violated (χ^2^(87) = 0.11, *p* < 0.001); therefore, the degrees of freedom were corrected using the Greenhouse–Geisser estimates of sphericity (ε = 0.46). Further analyses within the ROIs revealed that, in both time windows, this effect could be understood as a shift of the amplitude towards more positive values in the case of the question concerning competence than the question concerning warmth in both frontal regions. Specifically, it was expressed in the following way. In the 55–135 ms time window, in the **LF** region, the amplitude in the case of the question about competence (*M* = −0.76, *SEM* = 0.35) was less negative than for the question about warmth (*M* = −1.24, *SEM* = 0.39); *t*(29) = 3.03, *p* = 0.005, *d* = 1.11). Similarly, in the **RF** region, the amplitude in the case of the question about competence (*M* = −0.85, *SEM =* 0.39) was significantly less negative than in the case of the question about warmth (*M* = −1.30, *SEM =* 0.39; *t*(29) = 2.85, *p* = 0.008, *d* = 1.04).

Analogously, in the time window 135–175 ms, the post hoc tests revealed that, in the **LF** region, the amplitude in the “competence” condition (*M* = 2.38, *SEM =* 0.52) was more positive than in the “warmth” condition (*M* = 1.67, *SEM =* 0.60; *t*(29) = 2.65, *p* = 0.01, *d* = 0.97). Similarly, in the **RF** region, the amplitude in the “competence” condition (*M* = 2.14, *SEM =* 0.6) was more positive than in the “warmth” condition (*M* = 1.39, *SEM =* 0.64; *t*(29) = 3.16, *p* = 0.003, *d* = 1.15). The results are presented in Figure 6.

##### Time Window 315–500 ms

In the interval 315–500 ms, we obtained a significant main effect of the origin (*F*(1, 29) = 8.19, *p* = 0.008, *η**_p_**^2^* = 0.22). The amplitude was more positive in the Auto condition (*M* = 1.71, *SEM* = 0.31) than in Refl (*M* = 1.35, *SEM* = 0.3) (cf. Figure 7).

#### 3.2.3. Response-Locked Effects for Warmth–Competence Interpretation of Octagrams

When we analyzed the ERP data aligned with respect to the response, there was no good a priori baseline period for baseline correction. To overcome the problem, we analyzed the average time course of ERP for each condition without baseline correction (Appendix A) and found that, in the period from −600 to −500 ms, the traces for different levels of the analyzed factors were the most similar. Therefore, we used this time period as a baseline for the correction of epochs before further analysis. The ERP amplitude was analyzed in three time windows: −420 ms to −330 ms, −330 ms to −70 ms and −70 ms to 0 ms. The selection of these ranges was based on the GFP shown in Figure 8.

For this analysis, we selected nine EEG channels of interest: F3, Fz, F4, P3, Pz, P4, C3, Cz and C4, as we wanted to study the frontal, parietal and movement-related activity of the brain.

We performed a four-way ANOVA with repeated measures for each of the time windows. The complete statistical results, presenting ANOVA tables for all investigated factors and their interactions, are reported in Appendix A. Here, we describe only the statistically significant results.

The only significant effect we observed was in the period −420 ms to −330 ms. It was an interaction between the type of question and valence (*F*(2, 50) = 6.48, *p* = 0.003, *η**_p_**^2^* = 0.21). Further analyses for each type of question revealed a statistically significant main effect related to the levels of valence in the “warmth” condition (*F*(2, 50) = 5.52, *p* = 0.007, *η**_p_**^2^* = 0.18). Post hoc tests showed that the amplitude in the negative condition (*M* = 0.25, *SEM* = 0.26) was significantly more positive than in the neutral condition (*M* = −0.64, *SEM* = 0.2; *t*(25) = 3.14, *p* = 0.01, *d* = 1.23). Additionally, the analysis within valence levels revealed a statistically significant main effect in the case of the negative condition (*F*(2, 50) = 6.476, *p* = 0.003, *η**_p_**^2^*= 0.21). Post hoc tests showed that the amplitudes within the negative condition were significantly more positive for the question about warmth than about competence (*M* = −0.44, *SEM* = 0.22, *t*(25) = −2.82, *p* = 0.009, *d* = 1.11). The results are presented in Figure 9.

## 4. Discussion

In the present experiment, we aimed to study the role of two affective dimensions (valence and origin) on warmth and competence assessments in the ambiguous task paradigm, as well as to find the neural correlates of this process. We wanted to check whether we were able to replicate previous behavioral findings as well as find neural correlates for emotional stimuli, neutral stimuli and decision-making processes.

### 4.1. Behavioral Results

For the behavioral results, we partially confirmed our predictions: the stimuli of automatic origin elicited lower competence assessments compared to the stimuli of the reflective condition. This is particularly interesting, as this effect was not obtained in previous experiments [9,31]; the differences could be caused by the fact that, in the present study, we did not observe any interaction effect between the origin and valence (which occurred in previous experiments). Perhaps, we were able to study the main effect of origin without interlacing it with the valence effects, and thus we obtained different results because of the lack of disruption caused by the valence effects Nevertheless, it would be very much in line with the theoretical assumptions differentiating the relationships between the dualistic origin and social dimensions of warmth and competence [9].

Furthermore, in the present study, reflective words were causing a significantly higher competence than warmth assessments; this result is in accordance with previously proposed models of the duality of emotions [1,6] and previous experimental results [9,31] showing how deliberative, reflective processing is related to the competence dimension rather than warmth. However, there was no difference for assessments in warmth for words of different origin. This effect may be partially explained by the main effects that we obtained for the type of question (assessments performed on the dimension of warmth were, in general, significantly lower than assessments performed on the dimension of competence). It seems that, regardless of the origin of the presented words, in the ambiguous task situation, participants gave consistently similar warmth assessments. Possibly, the assessments of warmth—even in the uncertain evaluation of octagrams—were very automatic and intuitive, and the experimental manipulation would have to be stronger in order to involve reflective processing and to show differences on this dimension.

We also obtained interesting effects with the interaction of the type of question and valence. Although the assessment on both of the dimensions were increasing with the levels of valence (from negative to positive), changes between the levels of valence were more accentuated for warmth than competence. It seems very intuitive that the positive valence was tied up with higher assessments; however, it is interesting that this effect occurred not only for warmth, which is usually perceived as positive on a high level, but also for competence, which can actually be perceived as both positive and negative at its higher values. Nevertheless, in our study, these results were very similar for both warmth and competence (which can also be seen in the main effect for valence—in general assessments, they were higher for positive valence than negative or neutral) and show how the perception of social dimensions is tied to the emotional dimension of valence.

### 4.2. Word Reading ERP Correlates

The word reading part of the current paradigm may be treated as a typical emotional word-processing task. For the word reading and remembering task, we only obtained an interaction effect (with all the main effects being insignificant) between the ROI and origin. Namely, we observed that, in two ROIs (left-parietal and right-parietal), there was significantly more positive amplitude for reflective words than for automatic ones. A possible interpretation may be due to the fact that reflective words required more effort from the participants (as they may be more abstract and complicated, adhering to concepts and ideas; Refs. [1,6]) in processing than automatic words. This is an important result, as it occurred in the time window between 235 and 350 ms after the stimulus onset, therefore still being the second stage of word processing—the automatic one, which may be influenced by emotional factors [32]. It shows that the distinctiveness of the origin of words may be successfully observed in this stage in the early perception of the stimulus. This is especially interesting when we contrast this result with the fact that we did not obtain any result of valence, contrary to previous studies [46], and to the general definition of valence being the most basic dimension [24]. However, as in our study’s design, we crossed the levels of valence and origin and only obtained results of origin. Possibly, this is the dimension that may constitute the explanation (or a significant part of it) for the previous results.

### 4.3. Stimulus-Locked Analysis

The second analysis was conducted for neural correlates of neutral octagram inspection. The visual task was simple because the stimulus consisted of only black and white strips, but the aim of this visual inspection was to answer the task and to decide by intuition about the meaning of the stimuli; thus, we may expect the differences caused by this type of expectation. We observed that the type of task influenced two early time windows: 55–135 ms and 135–175 ms. The interaction between the type of question and ROIs revealed a shift of the amplitude towards more positive values when subjects were assessing competence, as compared to judgments of warmth. This effect appeared in both time windows and was localized in the two frontal regions—LF and RF. An interesting aspect of this effect is that functional magnetic resonance imagining (fMRI) studies have revealed the localization of such trait judgments to be the ventral part of the medial prefrontal cortex (vmPFC; [79]). However, this finding has been obtained separately for warmth [79,80] and competence [79], so comparisons between how the two trait judgments are neuronally realized are lacking. To our knowledge, this is the first observation of a difference in the early processing of visual judgments of warmth and competence for ambiguous stimuli, which suggests a difference in processing in the frontal regions that may be localized in the vmPFC.

Another interesting effect was the ERP modulation observed in the 315–500 ms window for words of different origin. Less negative amplitudes were observed when holding automatic words in memory when compared to reflective words. If we were to interpret this effect in the context of the “ERP Ambiguity Effects” literature [55,56,57], this effect could indicate a reduction in stimuli ambiguity for the automatic origin of emotional load. This could have, in turn, played a role in the observed behavioral effect of lower trait judgments when cued with automatic, compared to reflective, emotional load. Although we observed a change in trait judgments for both valence and origin manipulations, the appearance of a stimulus-locked modulation only for the origin may suggest that the mechanisms involved in these behavioral changes were different—with the origin influencing the way stimuli were perceived and valence influencing the decision process (as discussed in the following section). We did not observe any effect of valence in this analysis. However, when such effects are reported in the literature, a much stronger mood valence manipulation is employed compared to the emotional word remembering task of the present study.

### 4.4. Decision ERP Correlates

Finally, we analyzed the decision-locked potentials in order to investigate the decision-making process and decision execution. We revealed that the basic dimensions of social cognition were coming into statistical interaction with emotional valence when influencing the processing within the LRN. Assessing warmth in the condition of negative valence was related to a greater cognitive load than assessing warmth in a neutral condition and assessing competence in a negative condition. This is partially in line with our hypothesis; however, only the negative words were bringing large enough charge into processing to influence the ERPs. The effect was observed in the early stage of processing during the period of analyzing the stimuli and before initiating the response [63,64]. It was observed over the entire scalp and not in a particular region, which again confirms that the neural charge was not related to muscle activity [62].

This effect confirms the results of previous studies exploring the influence of the valence of LRP, in which the negative valence differed from other conditions [67,68]. Negative emotions seem to be more surprising in the context of a cognitive, experimental task conducted in a laboratory. In the study presented in this article, the warmth dimension seems to be in conflict with the negative condition; warm social traits are usually related to positive emotions, whereas competent ones could be interpreted in both positive and negative terms. Therefore, the negative words presented in the warmth condition were more surprising for participants than those in the competence condition. The same disturbance was responsible for the difference between the signals within the warmth condition between negative and neutral words, as the neutral ones were more in line with assessing warm traits than negative ones. We did not reveal any effects of origin within the LRN, which contradicts the results of our previous study [54]. Instead, in the present study, the effect of origin was observed as a modulation in the stimulus-locked analysis. This supports the claim that differences within the LRN are strongly related to the type of task used in the experiment [62].

### 4.5. Limitations

The current experiment has some limitations. The most important is that the experimental procedure was complex, based on two different stages. We were not able to create a single stage task, as is typical in EEG experiments, mostly because of the behavioral paradigm designed in earlier studies [1,9,31]. The three-stage analysis of ERP with different stimuli materials (words vs octagrams) did not allow us to cross-compare the effects inside the current experiment and limited the possibility of comparing the results with the literature. On the other hand, the division of the analysis into three stages provided more clear effects that can be more precisely controlled. Another limitation is the number of trials used in the factorial manipulations due to the inclusion of the question (type of fundamental dimension of social cognition) factor. In comparison to our other experiments investigating the valence and origin (cf. [9,81]), we decided to make a simpler experimental design, i.e., by resigning from mixed (unspecified) origin conditions. We decided to ask for warmth and competence separately, mainly because of the fact that both are separate in the social-cognition literature [10,11,14,15,16]. Additionally, we hoped to find the neural correlates specificity of assessing both those dimensions.

Finally, our limitation was also the sample size; however, in accordance with the a priori power analysis, it was still rather low. It is also important to remember that all of our participants were students, and thus, they were a rather homogenous group. Future studies should focus on replicating the effects obtained by us on both more numerous and various sample size.

## 5. Conclusions

The current experiment is the first demonstrating the association between the social-cognition duality of warmth and competence dimensions and emotional factors such as valence and origin. First, we identified the origin dimension to shape word processing during the first stage of the procedure (word remembering). We have also shown that the type of dimension assessed influenced the way participants inspected the visual octagram stimuli. At the same stage of analysis, but later in time, we found that the origin of emotional stimuli influenced the amplitude of potential after visual inspection (most likely when the process of guessing the semantical occurred). Finally, we found that the interaction of warmth and competence and emotional valence influences decision making in the LRN, but we did not find the corresponding behavioral data for the origin of emotional stimuli. The obtained pattern of results suggests that warmth and competence are processed in the mind slightly differently, and emotional factors may shape this processing.

## Figures and Tables

**Figure 1 brainsci-12-01093-f001:**
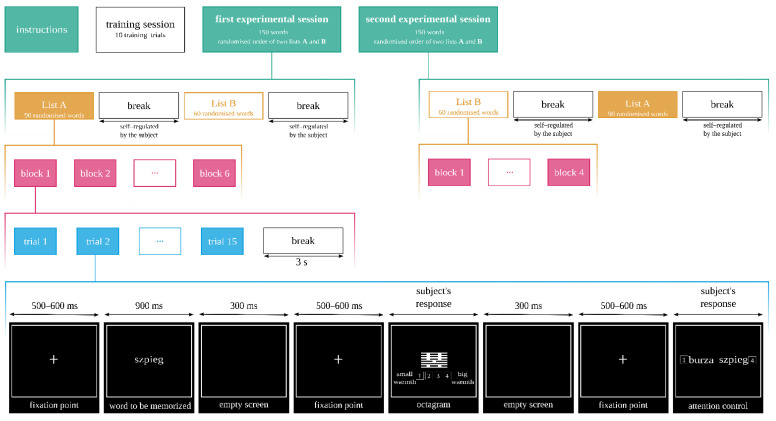
The illustration of the experimental procedure’s course: starting from the top, we show the timeline of the experiment, beginning with the instruction for the task, through randomized blocks of stimuli (lines 1–4 from the top)—conducted in two sessions. At the bottom line of the figure (black rectangles) we present an example of the single experimental trial with the fixation cross, example word stimuli (“szpieg” meaning “spy” in Polish), octagram presentation and attention control. Each screen of the trial was presented for the exact time written at the top of it.

**Figure 2 brainsci-12-01093-f002:**
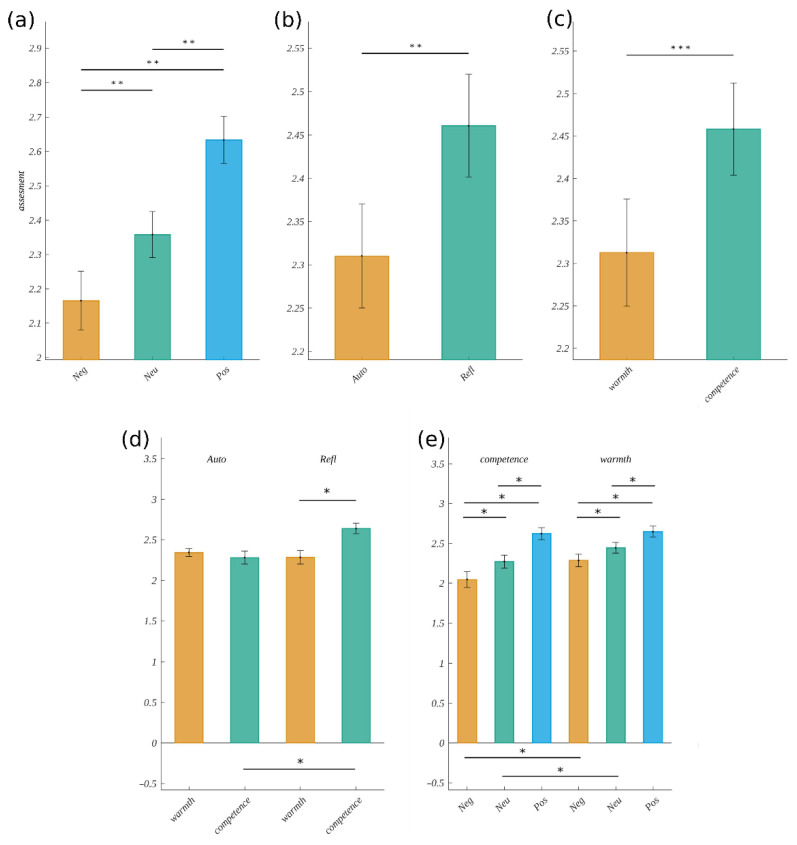
The assessment of how warm or how competent the octagram was (**a**) for different levels of valence, (**b**) for different levels of origin and (**c**) for the type of question. (**d**) The interaction effect of the emotional origin and the type of question, and (**e**) the interaction effect of valence and the type of question. The horizontal bar with the asterisks marks the significant difference (* *p* < 0.05, ** *p* < 0.01, *** *p* < 0.001).

**Figure 3 brainsci-12-01093-f003:**
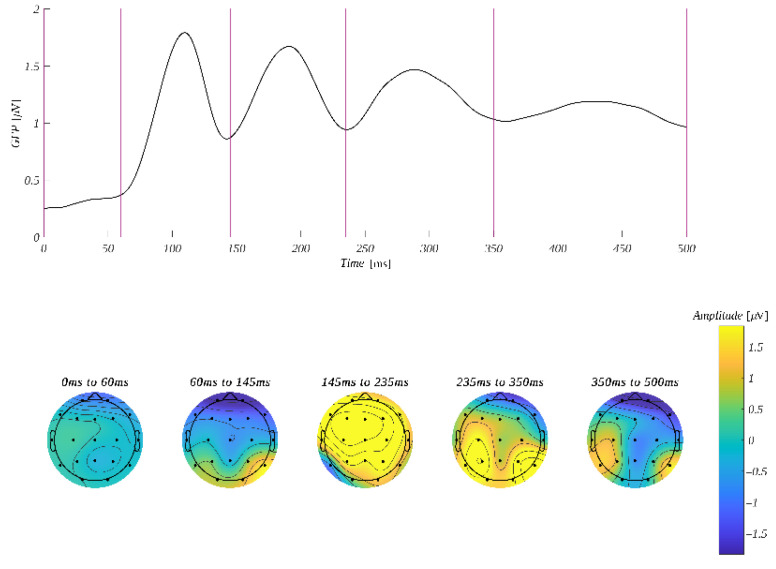
**Top**: GFP time course with 0 at the onset of the word reading task. Vertical lines mark the edges of the selected time windows. **Bottom**: average amplitude distribution within the selected time windows.

**Figure 4 brainsci-12-01093-f004:**
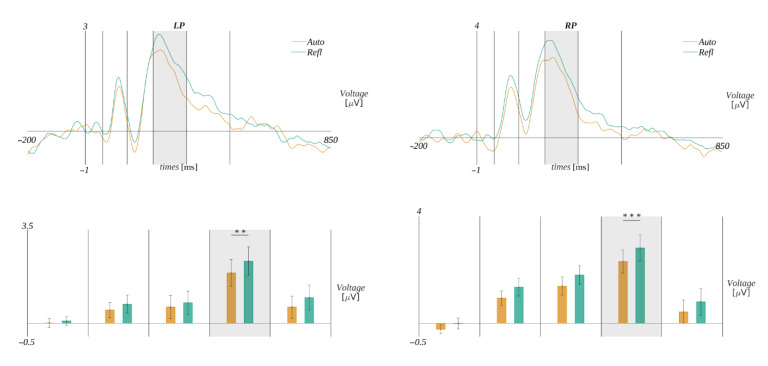
The ERP time course for the word reading task in the left parietal (**LP**) and right parietal (**RP**) region of interest. The grey rectangle marks the time window in which the difference between the amplitudes corresponding to automatic and reflective stimuli is significant. The bar plots below depict the mean amplitude (with *SEM* marked) within each time window for each of the conditions. The color code is according to the legends in the plots above. The horizontal bar with the asterisks marks the significant difference (** *p* < 0.01, *** *p* < 0.001).

**Figure 5 brainsci-12-01093-f005:**
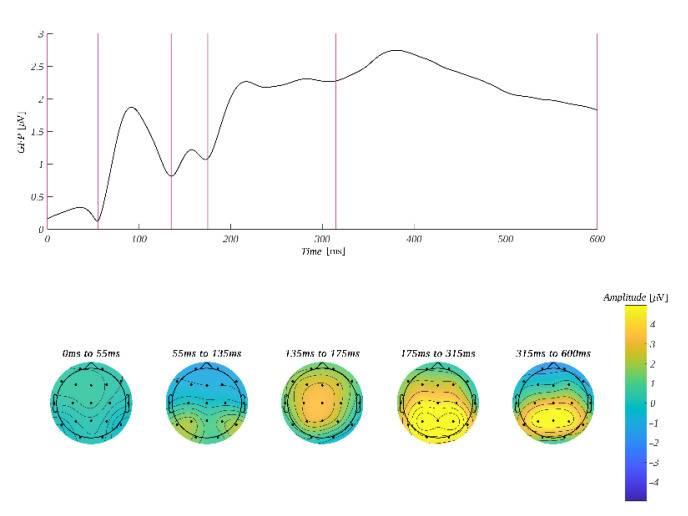
**Top**: GFP time course with 0 at the onset of octagram presentation. Vertical lines mark the edges of the selected time windows. **Bottom**: average amplitude distribution within the selected time windows.

**Figure 6 brainsci-12-01093-f006:**
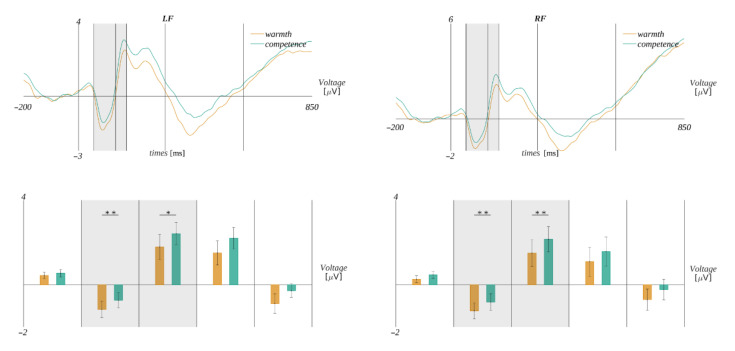
The illustration of the interaction effect between the type of question and regions of interest. **Top**: average ERP in the LF and RF regions observed for each type of question. Vertical lines mark the edges of the investigated time windows, and the gray rectangles mark the time intervals with significant differences. **Bottom**: bars indicate average amplitude within the given time window for the given condition. The color code is according to the legends in the plots above; error bars mark *SEM*. The horizontal lines with asterisks indicate the significance level (* *p* < 0.05, ** *p* < 0.01).

**Figure 7 brainsci-12-01093-f007:**
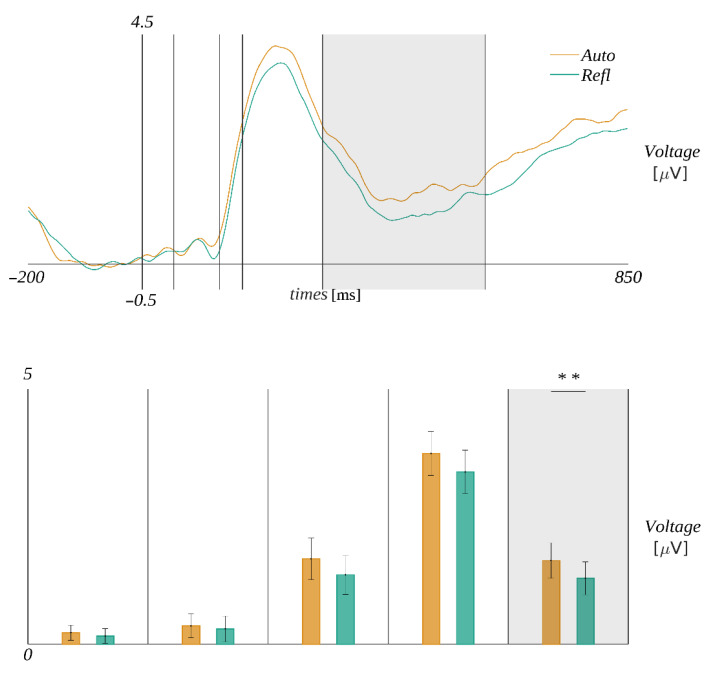
The ERP time course for the interpretation of the octagram task. The gray rectangle marks the time window in which the difference between the amplitudes corresponding to automatic and reflective stimuli is significant. The bar plots below depict the mean amplitude (with *SEM* marked) within each time window for each of the conditions. The color code is according to the legend in the plot above. The horizontal bar with asterisks marks the significant difference (** *p* < 0.01).

**Figure 8 brainsci-12-01093-f008:**
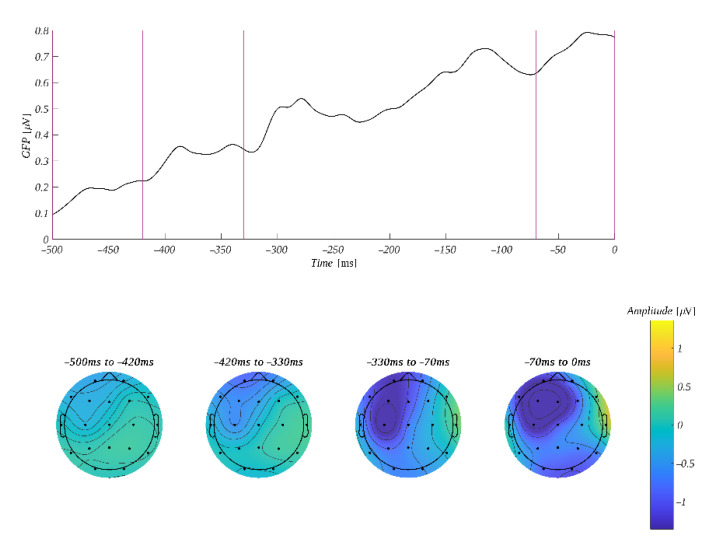
**Top**: GFP time course with 0 at the response key press. Vertical lines mark the edges of the selected time windows. **Bottom**: average amplitude distribution within the selected time windows.

**Figure 9 brainsci-12-01093-f009:**
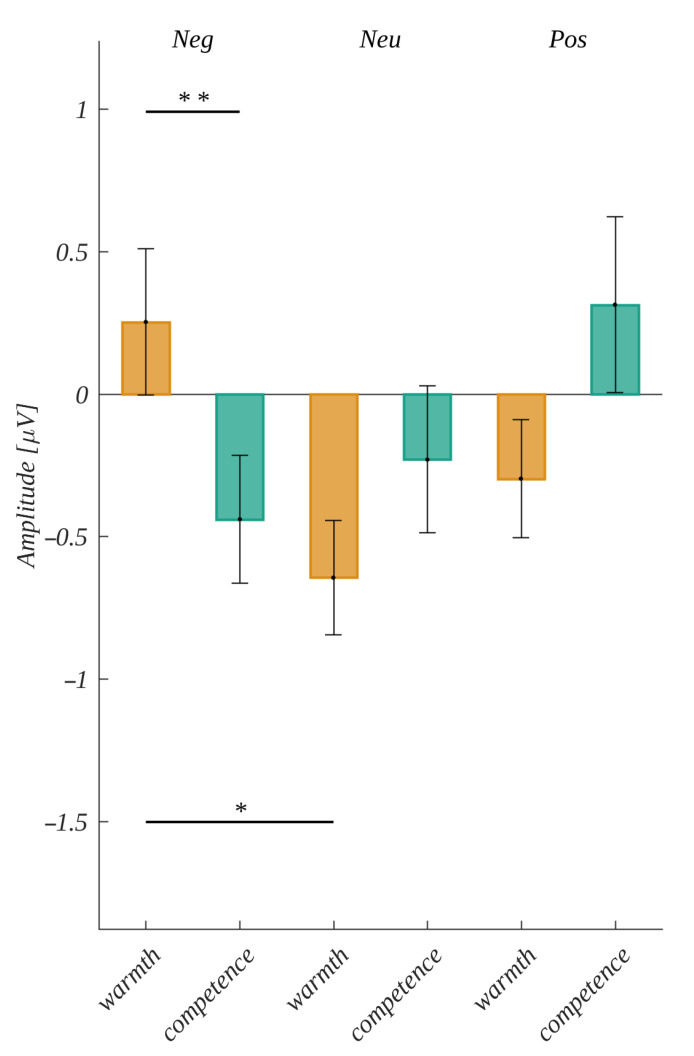
Interaction between the valence and the type of question for the period −420 ms to −330 ms before the response. The horizontal bar with the asterisks marks the significant difference (* *p* < 0.05, ** *p* < 0.01).

## Data Availability

The data presented in this study are openly available in the FigShare repository at https://doi.org/10.6084/m9.figshare.20263851.v1 (accessed on 9 August 2022).

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
