# Peer review of "The Affect Misattribution in the Interpretation of Ambiguous Stimuli in Terms of Warmth vs. Competence: Behavioral Phenomenon and Its Neural Correlates"

_brainsci, 2022, doi:10.3390/brainsci12081093_

Round 1
Reviewer 1 Report
In this paper, the authors probed the ability of factors such as valence and origin of an affective state to switch the interpretation of neutral objects. Behavioural results using octagrams show that negative valence and reflective origins promote the interpretation of objects in terms of competence. The authors also report event-locked EEG data and interaction of warmth, competence and emotional valence during the decision-making period. The experiments are executed well, and the data looks promising with exciting directions.
Author Response
Response to Reviewer 1 Comments:
In this paper, the authors probed the ability of factors such as valence and origin of an affective state to switch the interpretation of neutral objects. Behavioural results using octagrams show that negative valence and reflective origins promote the interpretation of objects in terms of competence. The authors also report event-locked EEG data and interaction of warmth, competence and emotional valence during the decision-making period. The experiments are executed well, and the data looks promising with exciting directions.
Response: Thank you very much for this review and the time that you took to read our manuscript.
Reviewer 2 Report
The goal of this study was to identify how the valence and origin modulate decisions of warmth and competence and how the valence of words influences the interpretation of ambiguous symbolic stimuli. The authors tested 36 participants in a double task: how the storeged emotion words task modulated the ambiguous stimuli evaluation task in term of warmth and competence. During the experiment, EEG was recorded to describe the ERP components correlated to the double task. The results suggest that emotional factors can modulate the processing of the big two traits, warmth and competence.
Before considering the paper of interest and useful to the readers of Brain Sciences, a minor review is necessary.
At line 77, the format of reference is wrong.
At line 119: "This stage is associated with modulation in early components, such as the early posterior negativity (EPN) or P300." I don't understand this sentence. The P300 is a positive component and the LPC that the authors describe, is part of P3 family.
At line 149, the author should include and discuss the following work:
Kissler J, Herbert C, Winkler I, Junghofer M.(2009) Emotion and attention in visual word processing: an ERP study. Biol Psychol. 2009 Jan;80(1):75-83. doi: 10.1016/j.biopsycho.2008.03.004. Epub 2008 Mar 14.
The caption of figure 1 should describe the image.
Author Response
Response to Reviewer 2 Comments:
The goal of this study was to identify how the valence and origin modulate decisions of warmth and competence and how the valence of words influences the interpretation of ambiguous symbolic stimuli. The authors tested 36 participants in a double task: how the storeged emotion words task modulated the ambiguous stimuli evaluation task in term of warmth and competence. During the experiment, EEG was recorded to describe the ERP components correlated to the double task. The results suggest that emotional factors can modulate the processing of the big two traits, warmth and competence.
Before considering the paper of interest and useful to the readers of Brain Sciences, a minor review is necessary.
At line 77, the format of reference is wrong.
RESP: Thank you for noticing that, we corrected this mistake.
At line 119: "This stage is associated with modulation in early components, such as the early posterior negativity (EPN) or P300." I don't understand this sentence. The P300 is a positive component and the LPC that the authors describe, is part of P3 family.
RESP: Thank you for noticing this. We deleted P300 from this sentence in order to make it more understandable.
At line 149, the author should include and discuss the following work:
Kissler J, Herbert C, Winkler I, Junghofer M.(2009) Emotion and attention in visual word processing: an ERP study. Biol Psychol. 2009 Jan;80(1):75-83. doi: 10.1016/j.biopsycho.2008.03.004. Epub 2008 Mar 14.
RESP: We included and shortly discussed the suggested work. Thank you for this recommendation, we found it really helpful and it allowed us to better make our point in the manuscript. We included this on page 7: “In a further investigation the differences in the N1 and EPN components were found, caused by the emotionality of the word stimuli and thus showing how the processing of valence in linguistic stimuli might be mostly found in early components [48].”
The caption of figure 1 should describe the image.
RESP: Thank you for this suggestion, it was especially important as the experimental procedure is quite complex. We added the description to the caption of the figure 1 and we hope that by that it is easier to understand and more clear for readers (page 17): “Figure 1. The illustration of the experimental procedure’s course: starting from the top, we show the timeline of the experiment, beginning with the instruction for the task, through randomized blocks of stimuli (lines 1 – 4 from the top) – conducted in two sessions. At the bottom line of the figure (black rectangles) we present an example of the single experimental trial with the fixation cross, example word stimuli (“szpieg” meaning “spy” in Polish), octagram presentation and attention control. Each screen of the trial was presented for the exact time written at the top of it.”
Thank you very much for your review, for the insightful and detailed comments as well as for the suggestion of relevant work to discuss in our manuscript. We hope that following all of your suggestions have improved our manuscript significantly.
Reviewer 3 Report
Dear authors
After reading your manuscript i have some suggestions:
1- please, explain the sampling method followed to obtain your sample.
Anda, please, include a brief explanation about rhe calculation to justify the 36 participants un your group.
2- It is imperative that hou nclude the information regarding to the approval by the Research Ethical Committe(REC). For example, i) the reference Code of the approval itself, ii) the university, hospital ir research center to which the REC belong.
3- The authors indicate that they don't collect any personal data but they do collect sexo or age from participants.
I suggest a modification of the text included on This section an eliminate This sentence.
4- please, could you explain why you decide use T-test and don't use correlational test like Pearson's test.
5- limitations. Please include the sampling method a and the number of participants
6- conclusions: I suggest eliminate the word 'link' (associated with causal effect) and use 'association'.
Author Response
Dear authors
After reading your manuscript i have some suggestions:
1- please, explain the sampling method followed to obtain your sample.
And a, please, include a brief explanation about rhe calculation to justify the 36 participants un your group.
RESP: Thank you for this suggestion. We added the sample size analysis which we conducted a priori (page 12): “We conducted an a priori estimation of our sample size. Basing on previous studies, we assumed that the ηp2 for main effects of one factor would be about 0.10. The estimations using G-Power software [69] showed that to achieve the statistical power of α = .80 we would need at least 20 participants. That small sample size is dictated by the specific study design – a significant number of repeated measures trials. However, we decided to increase the sample up to 36 participants in order to be able to conduct reliable interactions analyses and have a possibility to exclude any potential outliers.”
As for the recruitment, our participants were students of various faculties of Warsaw universities (also page 12).
2- It is imperative that hou nclude the information regarding to the approval by the Research Ethical Committe(REC). For example, i) the reference Code of the approval itself, ii) the university, hospital ir research center to which the REC belong.
RESP: Thank you for this comment, now we included the number of the approval (as an addition to the name the organisation granting the approval).
3- The authors indicate that they don't collect any personal data but they do collect sexo or age from participants.
I suggest a modification of the text included on This section an eliminate This sentence.
RESP: We deleted this sentence.
4- please, could you explain why you decide use T-test and don't use correlational test like Pearson's test.
RESP: Our choice of analysis method was caused by the design of the study: we wanted to compare different experimental conditions in a within-subject design. In our study we used t-test with appropriate correction for multiple comparison as a post hoc after finding a significant effect using ANOVA. This approach allows identifying which levels of independent variable are significantly different from each other. It also allows us to study the causal relationship and interpret our data accordingly.
5- limitations. Please include the sampling method a and the number of participants
RESP: We added this information to the limitation section, underlining the specifics of our participants: “Finally, our limitation was also the sample size – however in accordance with the a priori power analysis, it was still rather low. It is also important to remember that all of our participants were students, and thus they were a rather homogenous group. Future studies should focus on replicating the effects obtained by us on both more numerous and various sample size.”
6- conclusions: I suggest eliminate the word 'link' (associated with causal effect) and use 'association'.
RESP: We followed your suggestion and changed the wording.
Thank you very much for your review - we appreciate all the comments. We are really grateful for all of them, especially for detailed suggestions about the description of the sample size and the recruitment. We hope that following all of your suggestions have improved the manuscript significantly.